# Nasopharynx Battlefield: Cellular Immune Responses Mediated by Midkine in Nasopharyngeal Carcinoma and COVID-19

**DOI:** 10.3390/cancers15194850

**Published:** 2023-10-04

**Authors:** Ngar-Woon Kam, Cho-Yiu Lau, Chi-Ming Che, Victor Ho-Fun Lee

**Affiliations:** 1Department of Clinical Oncology, Centre of Cancer Medicine, School of Clinical Medicine, LKS Faculty of Medicine, The University of Hong Kong, Hong Kong 999077, China; yvonnekam@hksccb.hk (N.-W.K.); yoyocyl@hku.hk (C.-Y.L.); 2Laboratory for Synthetic Chemistry and Chemical Biology Ltd., Hong Kong Science Park, New Territories, Hong Kong 999077, China; cmche@hku.hk; 3Department of Chemistry, Faculty of Science, The University of Hong Kong, Hong Kong 999077, China; 4Clinical Oncology Center, The University of Hong Kong-Shenzhen Hospital, Shenzhen 518053, China

**Keywords:** midkine, ligand–receptor pairs, COVID-19, nasopharyngeal carcinoma, cellular immunity

## Abstract

**Simple Summary:**

One of the questions that experts are trying to answer about COVID-19 is whether it will have a positive or negative effect on cancer. It is important to consider the added risk of COVID-19 symptoms when making decisions about cancer treatments. To gain a deeper understanding of this, we discussed the relationship between COVID-19 and nasopharyngeal carcinoma (NPC), a type of cancer that, like COVID-19, initially causes inflammation in the nasopharynx. Specifically, we looked at the biomolecular variety and underlying intercellular ligand–receptor mechanisms through midkine (MK) signaling. We were astounded to discover how complex MK’s role is in the inflammatory response of these diseases, as it impacts the infiltration of immune cells into the inflamed microenvironment. This includes the abundance and interaction of immune cells, which can affect cancer progression and COVID-19 severity. As a result, exploring new cancer therapies that target MK gives hope of personalizing treatments for better outcomes.

**Abstract:**

Clinical evidence suggests that the severe respiratory illness coronavirus disease 2019 (COVID-19) is often associated with a cytokine storm that results in dysregulated immune responses. Prolonged COVID-19 positivity is thought to disproportionately affect cancer patients. With COVID-19 disrupting the delivery of cancer care, it is crucial to gain momentum and awareness of the mechanistic intersection between these two diseases. This review discusses the role of the cytokine midkine (MK) as an immunomodulator in patients with COVID-19 and nasopharyngeal carcinoma (NPC), both of which affect the nasal cavity. We conducted a review and analysis of immunocellular similarities and differences based on clinical studies, research articles, and published transcriptomic datasets. We specifically focused on ligand–receptor pairs that could be used to infer intercellular communication, as well as the current medications used for each disease, including NPC patients who have contracted COVID-19. Based on our findings, we recommend close monitoring of the MK axis to maintain the desirable effects of therapeutic regimens in fighting both NPC and COVID-19 infections.

## 1. Introduction

As of April 2023, the World Health Organization (WHO) had reported over 700 million confirmed global cases of severe acute respiratory syndrome coronavirus 2 (SARS-CoV-2) infections, the virus responsible for the COVID-19 pandemic [1]. The COVID-19 pandemic, caused by SARS-CoV-2, has resulted in a death toll exceeding 6 million people [2]. Scientists studying highly pathogenic coronaviruses have observed the hyperactivation of the host immune response to these viruses. Specifically, in the case of COVID-19, this uncontrolled immune response takes place in the nasopharynx, the area situated at the back of the nose and above the throat. As a consequence, it leads to severe lung damage and ultimately results in respiratory failure [3]. Moreover, research has shown that individuals with advanced cancer are at a heightened risk of developing severe stages of COVID-19 infection. It is noteworthy that patients with nasopharyngeal carcinoma (NPC) require particular attention as the tumor arises at the nasopharyngeal cavity, which is the primary site of SARS-CoV-2 entry and replication [4,5].

NPC is a metastasis-prone malignancy that is closely associated with the Epstein–Barr virus (EBV). EBV plays a role in early detection screening and also influences genetic and epigenetic changes in nasopharyngeal cells as well as immunomodulation [6,7]. Interestingly, a recent report suggests that severe cases of COVID-19 are associated with higher levels of EBV DNA [8]. However, unlike COVID-19, cancer development is typically associated with compromised immune responses. Cancer patients often exhibit an increase in the secretion of anti-inflammatory cytokines and a repression of the induction of pro-inflammatory mediators and cytokines that enhance the population of immunosuppressive leukocytes. Additionally, antineoplastic therapies can compromise the immune system and worsen immunosenescence. Consequently, cancer patients undergoing cytotoxic chemotherapy are at a heightened risk of developing infectious complications, such as COVID-19 [9].

The term “cytokine storm” was initially used to describe graft-versus-host disease. However, in recent years, this term has been extended to refer to the uncontrolled systemic overproduction of cytokines, which is observed in autoimmune conditions, sepsis, and other diseases, including COVID-19 infection and some cancers. While many cytokines are implicated in these uncontrolled systemic inflammatory responses, midkine (MK) has emerged as a key player in its pathophysiology and signaling pathways. MK is a secreted low molecular-weight growth factor (a heparin-binding cytokine) strongly involved in embryonic development [10]. Although the expression of MK is downregulated to relatively low levels in adulthood [11], it is induced and plays a role in tissue repair in situations where tissue damage occurs. Since its discovery, MK has been reported to be overexpressed in at least 20 different types of cancer [12]. Studies indicate that MK, or a combined biomarker test that includes MK, outperforms other current serum biomarkers in detecting malignancies at an early stage [13,14,15]. In COVID-19, high levels of MK have been linked to more severe illness and worse outcomes [16,17]. However, the current scientific literature is inadequate in identifying the specific immune subpopulations affected by MK in NPC and COVID-19 infections. Further research is needed to fully understand the role of MK in COVID-19 and NPC and explore its potential as a therapeutic target.

This article aims to comprehensively detail the MK-mediated cellular immunity that may impact immunosurveillance mechanisms in both of these nasal-related diseases. The review will begin via summarizing the physiological and pathophysiological properties of MK. We will then highlight the potential role of MK in contributing to carcinogenesis and cancer progression, with a specific focus on intercellular communication mediated by MK in the tumor microenvironment (TME) of NPC. The coordinated gene expression of ligands and receptors will be discussed, along with its implications for inferring intercellular communication, and new ideas for future application prospects will be put forward. In the context of the relationship between COVID-19 and NPC, possible cell–cell communication involving MK to regulate COVID-19 disease will be discussed. Additionally, current cases of NPC patients who have contracted COVID-19, leading to a significant interruption of cancer research, will be outlined. Finally, ongoing therapeutic strategies aimed at controlling the onset or damage caused by the overproduction of MK will be summarized, with the goal of reducing both morbidity and mortality.

## 2. Midkine Physiological and Pathophysiological Properties

MK, a 13 kDa secreted protein, was first discovered in the mid-gestation period of mouse embryogenesis, and in the adult mouse, the kidney was the principal site of its expression—hence the name “midkine” [18].

To date, significant advances have been made in elucidating MK-mediated mechanisms, which utilize numerous receptors and complicated intracellular signaling pathways to participate in a series of physiological and pathophysiological processes (Table 1). Proteoglycans, including receptor-like protein tyrosine phosphatase-ζ (PTPRZ1) [19], syndecans [20,21], and glypican-2 [22], demonstrate a strong affinity for MK. Additionally, other cell-surface non-proteoglycan receptors, such as low-density lipoprotein receptor-related protein (LRP) [23,24], anaplastic lymphoma kinase (ALK) [25], Notch2 [26], and several integrins [27], also serve as putative MK receptors. In general, the L-R interactions of MK are implicated in embryogenesis, organogenesis such as epithelio-mesenchymal organs [28] and neurogenesis [20], as well as cell growth and survival, which induce tissue regeneration processes, such as skeletal muscle regeneration [29] and cardiomyocytes repair [30].

The binding of MK to different receptors can activate several downstream signaling cascades. For instance, MK is essential for cell proliferation and the evasion of apoptosis during tissue repair and embryogenesis through PTPRZ1/syndecans/ALK/PI3K-related pathways [19,25,31]. Additionally, MK participates in regulating developmental and embryogenesis processes through stimulating the Notch2/Jak2-Stat3 pathway and inducing epithelial-to-mesenchymal transition (EMT) [26]. When MK interacts with the LRP-1 receptor on fibroblasts, it can contribute to the production of inflammatory mediators, such as cytokines (IL-6, IL-8) and chemokines (CCL2), which can activate immune cells enhance their infiltration [23]. Some studies have shown that MK can activate CD8+ T cells to produce CCL4, while others suggest that MK can induce CD8+ T cell dysfunction [32,33]. Furthermore, MK can activate CD4+ T cells to differentiate into T-helper 1 cells (Th1) [34]. Another study demonstrated that MK can suppress the development of tolerogenic dendritic cells (DCs) through STAT3 signaling, which can inhibit of regulatory T cell (Treg) differentiation [35]. In addition, the interaction between MK and LRP-1 has been found to triggers macrophage polarization towards type-2 phenotypes (M2) through enhancing the expression of arginase-1 [33]. Moreover, MK has been reported to contribute to the modulation of B cell survival through an MK-PTPRZ1-dependent pathway [36].

**Table 1 cancers-15-04850-t001:** Distribution, signaling network and biological responses of MK receptors.

MK-Binding Receptors	Receptor Location	Downstream Signaling Network	Association with Biology/Responses	Ref.
Non-immune cells				
Protein tyrosine phosphatase ζ (PTPRZ1)	Embryonic stem cell	PI3K/AKT signaling pathway	Embryonic stem cell self-renewal	[19]
Notch2	Keratinocytes	Notch2/Jak2/Stat3 signaling pathway	Cell plasticity and motility regulation	[26]
Syndecan-1 (SDC-1)	Neuronal cells	Neurite growth and neuronal cells migratory signaling	Early stage of neurogenesis	[20]
Syndecan-3 (SDC-3)			Development in post-natal period of neurogenesis	
Syndecan-4 (SDC-4, ryudocan)	Peripheral nerve bundles	Neurite promoting signaling	Formation of neural network	[21]
ALK	Fibroblasts, cancer cells, endothelial cells	PI3K and MAPK signaling	Cell growth, survival and angiogenesis	[25]
α_6_β_1_-integrins	Neurons	Growth signaling	Neurite outgrowth of embryonic neurons	[27]
α_4_β_1_- integrins	Osteoblastic cells	Migratory signaling	Haptotactic migration of osteoblastic cells	
Neuroglycan C	Oligodendrocyte precursor-like cell	Cytoskeletal modification, process extension and cell spreading	Neural network development	[37]
Low-density lipoprotein receptor-related protein (LRP-1)	Synovial fibroblasts	IL-6, IL-8 and CCL2 signaling pathway	Immune cells accumulation	[23]
	Fibroblast	Endocytosis of MK, nuclear targeting	Cell survival	[24]
Nucleolin (NCL)	Fibroblast cytoplasm	Nuclear translocation of MK		
Laminin binding protein precursor (LBP)	Rectal carcinoma cell	Nuclear translocation of MK	Tumor progression	[38]
Glypican-2	Neuronal cell	Neurite growth and neuronal cells migratory signaling	Neuronal development at embryonic stage	[22]
TSPAN1	Head and neck squamous cell carcinomas (HNSCC) cells	FAK phosphorylation and Stat1a pathway activation	Cell migration	[39]
NCAM	Embryonic cells	Unknown	Unknown	[40]
Brushin/megalin				
PG-M/versican	Embryonic cells	MK inhibition or promotion depending on the strength of binding	Unknown	[41]
LRP6	Embryonic neurons	Unknown	Neurons survival	[42]
apoER2				
Immune cells				
LRP-1	Polymorphonuclear neutrophils (PMN)	PMN recruitment by β2 integrin-dependent axis	PMN extravasation, trafficking and adhesion to inflammatory sites	[43,44]
	CD8+ T cells	Calcineurin/NFAT1 signaling	Naïve CD8+ T cells activation to produce CCL4, thus inducing microglia to produce CCL5	[32]
	Macrophages and tubular epithelial cells	MCP-1 pathway	Macrophage infiltration	[45]
	Macrophage	Increase M2 phenotype marker, arginase-1 expression	Alter microglia/macrophages polarization status	[46]
NCL	T cells	Formation of stable complex with NCL	Autocrine and paracrine regulate HIV infection	[47]
PTPRZ1	B cells	MIF/CD74 and HGF/c-Met signaling	B cells survival	[36]

## 3. MK Contributes to Carcinogenesis and Cancer Progression

Due to its multiple L-R interactions, it is not surprising that MK may be linked to a vast array of phenotypic characteristics that contribute to cancer development and progression. Studies have shown that ectopic expression of the MK gene in NIH3T3 cells can induce malignant transformation in mouse fibroblastic cells [48]. Several reports have suggested that tumors with higher levels of MK are more likely to be malignant in animal models of cancer. For instance, in non-small cell lung cancer (NSCLC) tumor-bearing nude mice, the administration of an MK inhibitor led to a significant reduction in neovascularization [49]. Similarly, inhibition of MK in mice engrafted with oral squamous cells resulted in suppressed tumor growth through the regulation of angiogenesis [50]. In mice with MK-knockdown-expressing liver cancer cells, the activation of the AMPKα pathway was observed to be involved in tumor cell survival and proliferation [51]. Additionally, Notch2, one of the MK receptors, was found to contribute to neuroblastoma (NB) tumorigenesis through Notch-HES1 signaling in an MK-deficient MYCN transgenic mice model for NB [52].

Due to its complex and dynamic nature, MK-induced inflammation can have various pro-tumorigenic consequences, such as angiogenesis [49,53,54], metastasis [49], mitogenesis [55,56], anti-apoptosis [57,58] and immune suppression [33,46,59] (Table 2). Interventions that can ablate MK expression have already proven to be beneficial. Specifically, administering the chemotherapeutic drug doxorubicin (DOX) in conjunction with a single-chain variable fragment against MK elicited an enhanced combinatorial benefit in tumor growth [60]. Furthermore, sorafenib treatment has been found to induce an immune-resistant state in an orthotopic hepatocellular carcinoma (HCC) mouse model. In this model, treatment with programmed cell death protein 1 (PD-1) blockade had no obvious inhibition on tumor growth, but the inhibitory effect was greatly enhanced through knocking down MK production induced by sorafenib [61]. Similarly, MK-expressing melanoma tumors increased resistance to anti-PD1 and anti-PD-L1 antibodies by which Tregs and F4/80 macrophages, rather than cytotoxic CD8+ T cells, were recruited into the tumor site [33]. This increased resistance to checkpoint blockade can be counteracted by a small molecule inhibitor of MK. Characterization of MK expression is therefore feasible for application in human cancer biomarker studies. In this manner, elevated serum levels of MK have been found to be associated with the presence of MK-expressing tumors. Previous studies have demonstrated that MK expression in serum/plasma and urine MK concentration are associated with various cancers, including esophageal, hepatocellular, colon, lung, breast, pancreatic, and oral squamous cell carcinoma. However, there is currently no evidence of MK expression in NPC (Figure 1).

### 3.1. Significance of MK in Tumor Microenvironment

It is now firmly established in cancer biology that TME, consisting of stromal fibroblasts, vascular cells, immune cells, and cancer cells, comprises an integrated network that contextually defines an individual tumor. Moreover, the abundance of immune cells plays a vital role in shaping the TME [73]. One example of MK-mediated signaling affecting the composition of the immune microenvironment in cancerous tissues was reported in inflammatory breast cancer. A CD151-dependent production of MK and its association with tumor cell-derived extracellular vesicles could regulate the chemotaxis of monocytes and macrophages to the TME [74]. We and others have reported that the TME, consisting of malignant and non-malignant cells (generally including stromal and immune cells), can generally be divided into two phenotypes based on the presence of tumor-infiltrating lymphocytes (TILs) [75,76]. Noninflamed tumors (cold tumors) are those with low infiltration of TILs, while inflamed tumors (hot tumors) are those with high infiltration and are therefore associated with better efficacy of immune checkpoint blockade (ICB) [77] and improved overall survival rates [78,79]. However, studies have highlighted that “hot” tumors may still fail to respond to ICB if the infiltrated cells are dysfunctional or become exhausted [80,81]. The feature of an inflamed but still immunosuppressive TME can potentially be fueled by various immunosuppressive cell populations, particularly tumor-associated macrophages (TAMs) [82,83], tumor-associated neutrophils (TANs) [84,85], myeloid-derived suppressor cells (MDSCs) [86,87] and Tregs [88,89]. Cerezo-Wallis et al. found that the secretion of MK from melanoma cells can boost an inflamed but immune refractory TME [33].

### 3.2. Intercellular Communication of MK in NPC

NPC is a malignant tumor derived from head and neck epithelial cells. The WHO has reported more than 133,000 new cases of nasopharyngeal cancer in 2020 [90], accounting for 1.6% of all cancer deaths, of which the mortality rate of male patients was 2.6 times that of females. It is known that NPC often reveals a large number of TIL in the primary tumor site [91]. Recent research has shown that in the process of the tumorigenesis, the composition and number of TILs subsets and their location in the TME of NPC are closely related to patients’ prognoses. Our group reported that majority of NPC patients presented with cold phenotype [75], while Wang et al. [92] showed that the amount of CD8+ TILs is positively correlated with survival in NPC. In principle, CD8+ TILs require CD4+ TILs to release cytokines to induce their proliferation. Thus, CD8+ TILs predicting a favorable prognosis may be due to the widespread infiltration of CD4+ TILs [93]. Although the CD4+/CD8+ ratios vary in different NPC specimens, they generally comprise over 50% of the TILs in NPC [94]. In addition, the recruitment of CD68+ monocytes and macrophage subsets into the tumor bed is usually associated with a poor prognosis, although this has not been shown to be statistically significant [95]. It has been found that LMP1, encoded by EBV, can induce a variety of cytokines and promote immune cells infiltration into the tumor site [96,97]. Meanwhile, the LMP1 protein is able to activate FoxP3+ Tregs to secrete the immunosuppressive cytokine IL-10 [98,99]. It can be seen that elevated expression of Tregs and CD68+ TAMs in EBV-positive NPC specimens can cooperate to promote tumor metastasis and are associated with a poor prognosis [100,101]. Thus, as an important part of the TME, TILs do not exist alone but as a complex multicellular population with high heterogeneity.

Lu et al. provided evidence of a direct link between miR-9 and MK activity in NPC through showing that miR-9 inhibits MK [102], and miR-9 has been found to be associated with the polarization of macrophages into an anti-tumoral M1-like phenotype in HPV+ cancer [103]. In line with this, MK overexpression can create an immunosuppressive TME in melanoma through attracting myeloid cells and M2-like TAM to the tumor site, which can weaken T cell activity [33]. Additionally, overexpression of MK through DNA-damage-induced p53 has been shown to remodel the immunosuppressive microenvironment in gliomas through promoting M2 polarization [46]. Therefore, it is conceivable that in NPC, the reduced levels of miR-9 [102] and increased expression of p53 [104] could potentially lead to increased expression of MK. This could weaken the miR-9 polarized M1-like effect and promote the accumulation of tumor-promoting (M2-like) macrophages in the tumor bed. Correspondingly, MK expression can promote the tumor infiltration of immunosuppressive MDSCs [61], a heterogeneous population of immature myeloid cells functionally defined by their ability to induce T cell suppression. In addition, MK can induce the production of IL-10 in tolerogenic DCs [35], extending the immunosuppressive TME through modulating the surrounding macrophages and T cells [100]. Furthermore, a high EBV burden in NPC can enhance IFN-γ production [105], and increasing IFN-γ has been found to promote MK secretion via STAT1 signaling in several epithelial cancer cell lines, such as lung and breast cancers [106]. It is conceivable that the high production of IFN-γ by EBV-NPC cells can help to promote a more enhanced expression of MK secretion, thus intensifying an inflamed yet immunosuppressive TME. Taken together, the discussed results propose that MK is a pro-tumorigenic modifier of the inflammatory milieu through maintaining an immunosuppressive TME and promoting tumor escape (Figure 2).

### 3.3. Ligand–Receptor Pairs in Disease in NPC

Single-cell RNA-seq (scRNA-seq) offers benefits over bulk analysis, particularly in quantifying expression in rare cell types and identifying the cell type of origin of proteins mediating cell–cell interactions. Here, we start via analyzing a publicly available sc-RNAseq database to explore cell–cell interactions between malignant cells and niche cell subtypes based on the L-R pairs. To infer potential L-R pairs that are likely used as means of intercellular communication, we applied a common single-cell inference method, CellChat [107], to two public scRNA-seq datasets containing NPC patient samples (GSE150430 and GSE162025; Figure 3). MK-expressing tumor cells are found to interact with 7 known receptors, including SDC1, SDC4, PTPRZ1, LRP-1, and the integrin family (ITGA4-ITGB1 and ITGA6-ITGB1) as well as NCL. To reveal the entire interaction state between cells, a communication score is assigned to the number of “active” L-R pair based on the RNA expression levels of their encoding genes in a given pair of sender and receiver cells. This score suggests which cells interact more strongly. In NPC, infiltrating immune cells were mainly contacted through the L-R pairs of MK-NCL in the MK signaling pathway. Such communication was also recently revealed in the work of Yu et al., where MK was considered to reconstruct an immunosuppressive environment in endometrial cancer [108]. MK was also found to support the progression of gastric cancer through interaction with NCL [109].

Unlike the previous report by Cohen et al. [36], we did not observe an interaction between MK-expressing NPC cancer cells and PTPRZ1-expressing B cells. Instead, the interaction was mediated by PTPRZ1-expressing CD45- non-immune stromal cells. Interestingly, MK-expressing tumor cells can interact with B cells and their subpopulation, plasma cells, through SDC1 and ITGA4-ITGB1, respectively. This type of interaction has not been detected before. Moreover, Guo et al. observed an MK-LRP-1 complex on CD8+ T cells in low-grade glioma [32], but this type of immune-cancer axis is not present in NPC. Therefore, although L-R pairs with MK-expressing tumor cells are not fully understood, we must be aware that the NPC microenvironment is unique and complicated. These findings shed light on the importance that we cannot directly adapt those developed therapeutics from other malignancies into NPC treatment.

To further delineate the relationship between MK expression and different immune infiltrates, we selected B cells, CD4+ T cells, Tregs, CD8+ T cells, and macrophages for correlation analysis. Consistent with the expected role of MK in contributing to macrophage accumulation, we observed that increasing MK levels correlated with elevated macrophage/monocyte abundance (r = 0.44 in GSE150430; r = 0.49 in GSE162025). In the NPC microenvironment, CD3+ T cells typically outnumber other immune subtypes [110]. Among CD3+ cells, we observed no correlation between MK levels and CD8+ T cells, but a negative, albeit moderate, correlation between CD4+ T cell abundance and MK expression in NPC cells (r = −0.44 in GSE150430). Studies have demonstrated that MK can inhibit Tregs differentiation in an autoimmune disease model [35]. However, a contradictory feature exists in the NPC immune landscape, where an increasing trend of MK-expressing tumor cells correlates with increasing Tregs (r = 0.36 in GSE150430; r = 0.58 in GSE162025). These findings highlight a novel aspect of MK function in NPC, namely its particular ability to modulate the infiltration of Tregs over other T cell lineages in the tumor microenvironment. Taken together, a hypothesis has emerged suggesting an MK-NCL-dependent immunosuppressive environment in NPC, which is associated with the accumulation of immunosuppressive macrophages/monocytes and Tregs in the TME. Ectopic overexpression of MK is thus implicated in the establishment of local immune tolerance in tumor tissues through “educating” immune cells via the MK-NCL signal.

## 4. Possible Contribution of MK to the Regulation of COVID-19

SARS-CoV-2 is a strain of coronavirus that causes COVID-19. Similar to all pathogens, SARS-CoV-2 uses various mechanisms to disable and evade the host immune system [111,112,113]. One of the presenting features of COVID-19 is hypoxia, and the induction of hypoxia-inducible factor 1-α (HIF1-α) is considered an important component of SARS-CoV-2 infection [114]. Increased MK expression has been reported during hypoxia, as HIF1-α binds to hypoxia-responsive elements located in the MK promoter [115], suggesting that the hypoxia-driven inflammatory misalignment by MK may contribute to COVID-19. Another important issue with COVID-19 is the development of acute respiratory distress syndrome (ARDS), which can cause diffuse alveolar damage in the lungs and even death [116]. Several studies have demonstrated that MK plays a role in the pathogenesis of ARDS. For instance, when MK expression was silenced, a reduced EMT profile in lung epithelial cells was observed [117]. Similarly, a higher MK level was detected in the serum of patients with idiopathic pulmonary fibrosis patients compared to healthy subjects, supporting the role of MK in the development of ARDS-associated in COVID-19 [118]. Supporting this interpretation, serum MK is significantly upregulated in COVID-19 patients compared to normal individuals, with mean values of 1892.8 ± 1615.8 pg/mL and 680.7 ± 907.6 pg/mL, respectively [16].

Early and moderate COVID-19 has been found to remain in the upper respiratory tract, eliciting a minimal innate immune response [119]. However, if left unchecked, the immune response can result in immunopathological changes. The non-specific innate immune response is followed by production of proinflammatory cytokines such as IL-6, and the recruitment of neutrophils and myeloid cells to the inflamed region [120]. Although SARS-CoV-2 infection in host cells elicits robust secretion of chemokines and cytokines, impaired type 1 interferon (IFN) immunity is identified in COVID-19 patients [121]. On the other hand, cytokine-induced alterations in COVID-19 are accompanied by excessive production of cytokine inflammatory mediators (IL-1β, IL-6, TNF-α) [122]. TNF-α was observed to induce the expression of MK [23,58]. Therefore, it is likely that MK is closely involved in cytokine invasion and interacts with other inflammatory factors during SARS-CoV-2 infections.

### 4.1. Intercellular Communication of MK in COVID-19

Luo et al. observed that the spleen and lymph nodes of six COVID-19 patients who died contained CD68+ and CD169+ macrophages expressing the ACE2-SARS-CoV-2 complex, indicating the significant impact of macrophages on viral spread during COVID-19 [110]. Recently, it was reported that severe COVID-19 patients had significantly increased levels of highly inflammatory monocyte-derived FCN1+ macrophages and CD14+ CD16+ inflammatory monocytes, while FABP4+ alveolar macrophages were greatly reduced in their bronchoalveolar lavage fluid (BALF) [123,124]. Neutrophils also play a crucial role in this myeloid-driven immunopathology. Elevated levels of Neutrophil Extracellular Traps (NETs), also known as NETosis, were found in the blood, thrombi, and lungs of severe COVID-19 patients, suggesting the essential role of neutrophils in initiating lung damage during COVID-19 [125,126].

In several inflammatory diseases, MK has been shown to exacerbate the condition through promoting NETosis, neutrophil trafficking, and macrophage accumulation [43,44,127,128]. Based on the findings discussed in the preceding section, it is reasonable to suggest that accelerated MK expression contributes to the recruitment of innate immune cells, which helps to modulate the immunopathology in COVID-19 patients. Moreover, it is important to note that the M1-M2 phenotypic spectrum in COVID-19 is complex. Studies show that both M1/pro-inflammatory or M2/anti-inflammatory macrophages have comparable abilities to eliminate the virus in the context of a moderate viral load. However, the M1 type has been found to cause more harm to lung cells and release a set of inflammatory factors [129]. As such, the distinct immune response patterns of macrophages may partially account for the diverse susceptibility and symptoms of COVID-19 observed among individuals.

Following the non-specific innate immune response, antigen-specific adaptive immunity mediated by B cells (humoral immunity) producing neutralizing antibodies and T cells (cellular immunity) including CD8+, CD4+, and Tregs is activated. A higher neutrophil-to-CD8 ratio has been considered a predictor of a severe COVID-19 course [130,131], while increased CD4-to-CD8 T cell ratio serves as a good prognostic marker for disease severity with mortality prediction power [131]. SARS-CoV-2-specific CD8+ T cell responses have been detected in the acute and convalescent phases of COVID-19 [132,133,134,135,136,137], and CD8+ T cells have been found to contribute to protection from the development of severe COVID-19 in animals experimentally infected with SARS-CoV-2 [138,139]. However, severe COVID-19 is characterized by functional exhaustion and decreased numbers of T lymphocytes [140,141]. The impaired T cell responses are likely due to the deficient IFNs production driven by SARS-CoV-2, since IFNs are used to promote the survival and effector functions of T cells. Xu et al. reported elevated frequencies of CD4+ T cells among all CD3+ T lymphocytes in COVID-19 disease [142], which was consistent with earlier reports suggesting that lymphopenia affects CD8+ T cells more than CD4+ T cells [143]. Zhao et al. also found a higher trend of clonal expansion in CD4+ T cell subsets [144]. Therefore, we suspect that the disturbed T cell compartments could play a role in COVID-19 immunopathogenesis, leading to impaired antiviral responses by CD8+ T cells and exacerbated inflammation caused by abnormal B cell-mediated antibody responses by CD4+ T cells. MK has been shown to regulate and interfere with the activity of both CD8+ and CD4+ T cells [32], indicating a possible role for MK in the progression of COVID-19. Unlike CD4+ T cells, which are known to be activated by MK [34], there are conflicting reports on the effect of MK on CD8+ T cells, with some studies showing an inhibitory effect [32] and others showing an activating effect [33]. Nevertheless, these findings suggest that MK may play an intermediary role in the phased progression of COVID-19.

### 4.2. Ligand–Receptor Pairs in Disease in COVID-19

MK elevation has been demonstrated in COVID-19 [16], but the interaction between the MK signal pathway and surrounding cells has not been well documented to our knowledge. Considering the diverse receptors recognized by MK and their associated consequences, including lymphopenia, aberrant granulocytes and monocytes, cytokine storm, and an increased neutrophil-to-lymphocyte ratio (NLR), which are all correlated with disease severity [145,146,147,148], we will discuss potential L-R pairs based on a public sc-seq dataset of patients with COVID-19 (GSE145926) in the following subsection.

MK exhibits a restricted expression pattern, mainly on epithelial cells followed by macrophages. While MK-PTPRZ1 signaling is not present in the microenvironment of BALF in comparison to NPC, the MDK-NCL complex still remains the top L-R pair across all cell pair interactions. Additionally, the previously unrecognized L-R pair involving SDC1 and ITGA4-ITGB1 can also be observed between MK-expressing epithelial cells and B cells. It is possible that the MK-dependent biological function of B cells, i.e., MK-SDC1 or MK/ITGA4-ITGB1, might be exclusively present in the nasal cavity to mediate any possible inflammatory responses in malignancy and SARS-CoV-2 infection.

As previously mentioned, COVID-19 patients exhibit a characteristically high CD4:CD8 ratio [143], and there is evidence suggesting that MK regulates T cell activities [32,33]. We suggest that MK plays a role in relation to lymphocyte frequencies, particularly the T cell compartment, during the course of SARS-CoV-2 infection. As illustrated in Figure 4, the expression levels of MK were inversely correlated with the abundance of CD8+ T cells (r = −0.5) and positively correlated with CD4+ T cells (r = 0.58). Such a correlation is not observed in the TME of NPC. It is worth noting that CD8+ T cells are crucial for attacking and killing virus-infected cells directly, while CD4+ T cells are important in priming both CD8+ T cells and B cells. Additionally, a significant reduction in CD8+ T cells has been observed in peripheral blood in patients who died from COVID-19 [149]. These findings strongly suggest that MK signaling mechanisms may play a role in either CD8+ T cell depletion or CD4+ T cell enrichment, mediating different biological effects that contribute to the development of “long COVID-19”. In addition, although a positive correlation was found between MK-expressing NPC cells and Tregs, no association was observed for MK-expressing epithelial cells and Tregs in COVID-19. Masuda et al. have previously reported the involvement of MK in the Treg-independent differentiation into Th1 cells in a lupus-prone mice model [34]. Thus, the balance between differentiation into Th1 effector cells or Tregs by MK might be somewhat different in chronic infections, autoimmune diseases, and certain cancers. If these considerations prove to be true, it is highly possible that MK selectively modulates the CD4+ T cell population based on the surrounding inflammatory conditions, with a trend towards a higher frequency of Tregs in the immunosuppressive TME and a higher frequency of Th1 during hyperinflammatory microenvironments, such as COVID-19 disease.

Regarding myeloid populations, MK expression was found to positively correlate with macrophage abundance (r = 0.35), which is consistent with the observations made in the NPC cohort. Interestingly, despite the high number of neutrophils found in BALF, they did not show any L-R interaction with MK signaling. It has been reported that COVID-19 severity is associated with an elevated MK serum level [17] and an increased NLR ratio [130,131,150]. Furthermore, experiments conducted in MK knockout mice have shown that neutrophil infiltration and chemokines, such as macrophage inflammatory protein-2 (MIP-2) and monocyte chemoattractant protein-1 (MCP-1), were reduced in tubular epithelial cells [151]. Moreover, MK is a critical factor that links LRP-1 to β2 integrin function for trafficking neutrophils during acute inflammation [44]. Therefore, considering that MK is involved in the recruitment of inflammatory cells, one can easily hypothesize that MK might indirectly contribute to neutrophil migration through the induction of chemokines, such as MCP-1 and MIP-2, in progressing to severe COVID-19. Additionally, there is a possibility that other undefined receptors on neutrophils, which are not associated with acute inflammation, could interact with MK when COVID-19 progresses from mild to severe signatures, as seen in Figure 5.

## 5. Impact of COVID-19 on Outcomes for Patients with NPC

### 5.1. Managing NPC and COVID-19: Challenges and Clinical Evidence for Standard Care Therapies

Comparable immune dysregulation caused by malignancies and COVID-19 support our hypothesis for a cellular-immunologic interplay between COVID-19 and cancer progression (Figure 5). In this section, several patient cases presenting a disproportionate imbalance in immunity will be discussed, illustrating the significant challenges encountered by oncology practitioners when faced with the decision of treating a cancer patient with an ongoing antineoplastic treatment who has also been diagnosed with COVID-19.

In 2020, a case study [152] revealed that a patient diagnosed with NPC developed a fever on the third day after undergoing combination treatment with sintilimab, an anti-PD1 antibody, and chemotherapy. Despite the white blood cell counts remaining normal, increased neutrophil-to-lymphocyte and CD4/CD8 T cell count ratios were observed, along with elevated levels of IL-6 and IL-10. Similarly, a case study of a 51-year-old man [153], diagnosed with NPC and treated with a combination of radiotherapy and chemotherapy, showed a consistent reduction in total lymphocyte and T cell counts. The patient tested positive for COVID-19 immediately after completing radio-chemotherapy. Additionally, another case report discussed a 14-year-old patient with COVID-19 who was previously diagnosed with stage IV NPC and treated with a series of radio-chemotherapy [154]. The patients took two weeks of intensive care before tested negative.

Despite the prevailing notion that patients with comorbidities like cancer and post-chemoradiation have weakened immunity, which can facilitate the transmission of coronavirus infection in these patients, Venkatraman et al. proposed the use of radiotherapy as a therapeutic approach in COVID-19 cases. They suggest that low-dose radiation may be effective in killing the virus [155]. Additionally, Hua et al. reported that NPC patients treated with anti-PD-1 and chemotherapy could benefit from COVID-19 vaccination [156]. This study demonstrated that NPC patients vaccinated with SinoVac responded significantly better to anti-PD-1 therapy than unvaccinated patients. On the other hand, a prospective cohort from Lee et al. argued that the presence of cancer and the receipt of cytotoxic anti-cancer treatment did not affect the COVID-19 disease phenotype [157]. This group reported no significant differences in the mortality rates in COVID-19 patients with cancer who received or did not receive chemotherapy/radiotherapy within four weeks of testing positive for COVID-19. However, it remains unclear why COVID-19 vaccination supports immunotherapy treatment and why NPC patients are at a high risk of developing COVID-19 after immunotherapy.

### 5.2. The Potential Influence of MK-Mediated Cellular Immunity on COVID-19-Contracted NPC Cancer Progression and Therapy Response

Our investigation of public cohorts has narrowed down the potential interaction between MK and immune infiltrates in both COVID-19 and NPC patients. However, it remains an open-ended question whether MK overexpression could reduce the success of cancer treatment or be incompatible with vaccination. In supporting the former scenario, a current study described that high MK expression in mouse melanomas can trigger resistance to immune checkpoint blockade strategies through targeting PD-1/PD-L1, whereas the ablation of MK can re-sensitize cancers to the immunotherapies [33]. The inability of CD8+ T cells to reach tumor cells [158] and the induction of T cell exhaustion [159] are both considered as significant parameters for resistance to cancer immunotherapy, whereas increased T cell exhaustion, reduced functional diversity and decreased in CD8+ T cell values are reported to be indicators of severe COVID-19 diseases [141,160]. It is conceivable that cancer cells overexpressed with MK may impede the reactivation of exhausted CD8+ T cells in the TME during COVID-19 vaccination and facilitate the tumor’s escape from immune surveillance. This, in turn, may lead to the development of high rates of inflammation and poor COVID-19 disease progression. As mentioned previously, MK expression in NPC cells negatively impacted CD8 T cell abundance; it is therefore possible that MK-induced dysregulation of the adaptive immune system, such as a reduction in CD8+ T cell frequencies, can lead to a greater severity for COVID-19 in patients with NPC. In addition, the inadequate CD8 T cell counts may partially transfer to the results observed increased CD4/CD8 ratio in patients with COVID-19 [160]. Furthermore, Hu et al. demonstrated MK regulates the drug efflux pump [161], while Gungor et al. described that pancreatic tumors depleted of MK become susceptible to chemotherapy [162]. Moreover, MK is involved in sculpting the macrophage phenotype and driving immune cells chemotaxis, as discussed above. These findings indicate that abnormal expression of MK is considered to be an important molecular mechanism in the obstruction of immunotherapeutic responses, possibly through the alternation of CD8+ T cell and macrophage population, and also one of the causes of drug resistance in cancer patients complicated with COVID-19.

## 6. Implications for Therapies

Multiple reports have shown that patients with malignancy can have particularly adverse outcomes with COVID-19 [163,164,165,166,167,168,169]. It is worth noting that nasopharyngeal cavity is a critical stop for the transmission of respiratory viruses to the lungs. Meanwhile, the nasopharyngeal cavity is a primary reservoir of SARS-CoV-2 [4,5]. Furthermore, radiotherapy and chemotherapy can damage the epithelium of the oral mucosa, compromising the normal barrier structure [170] and complicating the immune system. All these factors increase the risk of infection among NPC patients.

### Clinical Benefits of Therapies against MK Signaling

Given that the MK-NCL axis has been identified as a significant L-R pair in both immunosuppressive NPC and hyperinflammatory COVID-19, the pharmacological targeting of MK-NCL signaling may be a promising approach to reshape the local immune microenvironment to be more therapy-permissive. This would aim to enable a balance of cellular immunity between the risks and benefits of cancer-directed interventions within the context of the added risk of SARS-CoV-2 infection.

As the function of MK in immune cellularity is increasingly appreciated, we hypothesize that targeting MK could be a therapeutic approach to stabilize inflammatory states, restore the balance of immune cells, and enhance the efficacy of treatment regimens. Currently, MK inhibitors, including antibodies, aptamers, glycosaminoglycans (GAGs), peptides, and low-molecular-weight compounds, are in the preclinical development stages [171]. Moreover, in a study by Takei et al., MK gene knockdown using small interfering RNA (siRNA) in combination with paclitaxel treatment was found to suppress tumor growth in a xenograft model of prostate cancer [172]. Additionally, the siRNA-mediated inhibition of MK expression and antisense MK oligodeoxyribonucleotides have shown therapeutic antitumor activity [173,174,175,176]. Furthermore, MK has been recognized as a tumor antigen for cancer vaccine and gene therapy development. Evidence has shown that the immunogenic sequences (T cell epitopes) of CD4 [177] and CD8 [178] reside in the signal peptide of MK, highlighting the involvement of MK in specific anti-tumor responses. In other trials, an MK promoter-based conditionally replicating adenovirus has been proposed for treating cancers overexpressing MK [179,180]. The local replication of oncolytic virus can induce specific antitumor immunity during its oncolytic activities. This MK-based therapy has been established in pancreatic cancer [181], glioma [182], and HNSCC [183].

Given the wide range of therapeutic applications, it is not surprising that numerous patents have been filed for MK drugs against various cancers (Table 3). However, the therapeutic potential of MK in NPC remains undefined. We speculate that inhibiting or blocking MK’s mode of action prior to or during chemotherapy or immunotherapy may force immuno/chemo-resistant cells to revert to a more sensitive state, providing a second opportunity for NPC patients who are unresponsive to conventional treatments or immunotherapeutic strategies. Moreover, we believe that the increased susceptibility of cancer patients to SARS-CoV-2 infection may be due to higher baseline inflammation, either from the disease itself or the therapies used to treat it, which is then aggravated by the infectious disease. Therefore, targeting MK may also have benefits in countering COVID-19 hyperinflammation through attenuating systemic inflammation and thus preventing synergistic inflammation.

From a drug development perspective, one direction adopted by pharmaceutical companies is to commercialize anti-MK drugs for the treatment of both COVID-19 and cancers. A UK-based biotech company is investing a large sum of money in MK research to develop blocking antibodies, oligonucleotides, and cellular therapeutics. They foresee market potential for MK in COVID-19, cancers, chronic inflammation, and autoimmune diseases. According to the company’s website, two patent-protected antibodies that specifically target cancer metastasis have been validated in animal models, followed by the successful completion of GLP toxicology studies. Additionally, a novel platform has recently been established for four mRNA therapeutics targeting MK, which are now in the pre-clinical stage [184].

## 7. Conclusions

To summarize, COVID-19 and cancer, despite having discrete etiologies, both involve an imbalance in the immune cell compartment. The depletion of CD8 T cells, which are essential for immune response, by COVID-19 may aggravate cancer progression in NPC patients, suggesting a potentially negative impact. Furthermore, the interaction between immunomodulatory molecules and immune cells during COVID-19 infection may further disturb the already-compromised immune system in NPC, exacerbating the negative effects. On the other hand, in some cases, if COVID-19 triggers a robust immune response in NPC patients, it may lead to improved tumor surveillance and the enhanced immune recognition of cancer cells. This introduces uncertainties regarding the directional effects of COVID-19 on NPC, as the impact may vary depending on different contexts. As discussed in this review, MK is involved in a multitude of biological and cell–cell communication processes and is believed to play parallel roles in the mechanisms of immunosuppressive cancers and the COVID-19 cytokine storm. As such, MK could be a useful “soldier” to fight against the early phases of cancers and COVID-19 infections, which are dominated by the innate immune response. However, as the diseases progress, MK may become a driving force of dysregulation in the local inflammatory milieu. This highlights the complex and context-dependent nature of the interplay between COVID-19, cancer, and the role of specific immunomodulatory molecules like MK.

NPC is an immune hot tumor strongly associated with EBV. The unique immune environment presents rational targets for immunotherapy. While adoptive cell therapy and therapeutic vaccines have shown initial success in treating a small subset of refractory NPC patients, the clinical translation of immunomodulators and EBV-specific monoclonal antibodies against NPC is still in the exploratory phase. It is worth noting that EBV-specific vaccines can induce specific antitumor responses through directly or indirectly targeting tumor antigens. However, these EBV-specific agents have limited targets and are costly and technically immature to produce. Therefore, exploring the molecular and cellular drivers of immune escape in NPC may lead to innovative therapeutic options. These options can not only improve outcomes for NPC patients but also simultaneously manage the multiple comorbidities associated with NPC patients who have contracted a COVID-19 infection.

Overall, therapeutic strategies targeting MK, either alone or in combination with the current standard of care, are needed for personalized medicine. Further research is needed to fully understand the biological role of MK in driving carcinogenesis and therapeutic interventions in patients with NPC who develop COVID-19. Finally, the cell–cell communication data in this review were obtained from limited samples, and thus future work based on larger datasets from NPC and COVID-19 patients is necessary to generalize our findings. In addition, our hypothetical ligand–receptor pairs are mainly based on published transcriptomic data, and further functional experiments are required for validation.

## Figures and Tables

**Figure 1 cancers-15-04850-f001:**
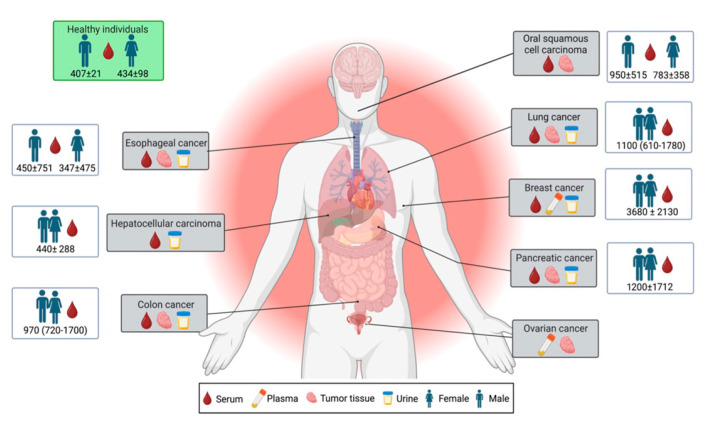
Analysis of MK expression in human cancer samples. Serum levels of MK in healthy individuals (box in green) and various human cancers (open box) are indicated. Values are represented in pg/mL. Esophageal carcinoma [62]: mean ± 2 SD; healthy control [63], hepatocellular carcinoma [64], oral squamous cell carcinoma [63], breast [65] and pancreatic cancer [66]: mean ± SD; colon [67] and lung cancer [68]: median (25–75 percentile).

**Figure 2 cancers-15-04850-f002:**
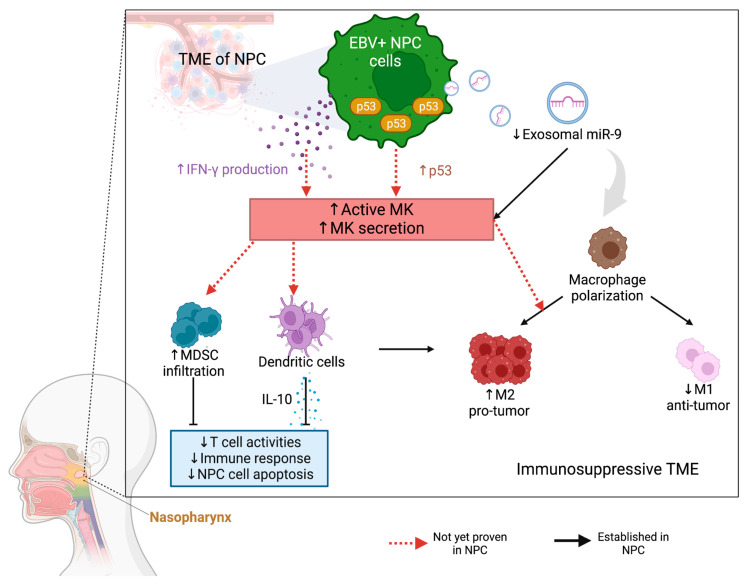
MK-mediated aberrations in key cell composition promote immunosuppressive NPC microenvironments. Dysregulation of cytokines, regulatory protein, and exosomes by altered EBV+ NPC cells can either increase or decrease immune cell accumulation, thereby shaping the TME of NPC. Arrows denote activation and blunted lines indicate inhibition. Red dashed arrows designate the proposed mechanisms that remain to be studied further.

**Figure 3 cancers-15-04850-f003:**
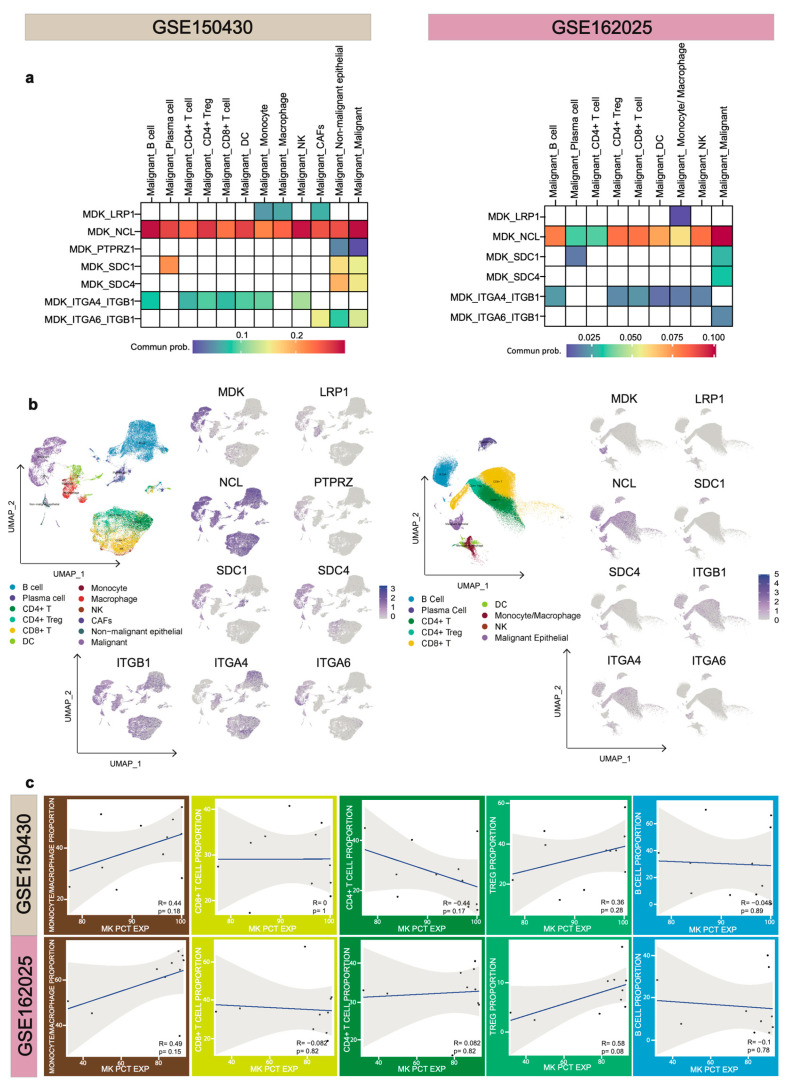
Analysis of two NPC scRNA-seq cohorts. (**a**) Heatmap presents the ligand–receptor communication probability in NPC tissues (GSE150430, n = 11; GSE162025, n = 10). Communication probability was visualized using the ggplot package. (**b**) Expression of cell type clustering (**left**) and ligand–receptor expression (**right**), presented as expression threshold values (key), overlaid on the t-SNE maps. (**c**) MK percentage expression (MK PCT EXP) (*x*-axis) plotted against the cell type abundance (*y*-axis), presented as a scatter plot: blue solid line, linear regression (least-square method), and Pearson correlation coefficient ‘R’ are displayed. Each dot represents an individual patient.

**Figure 4 cancers-15-04850-f004:**
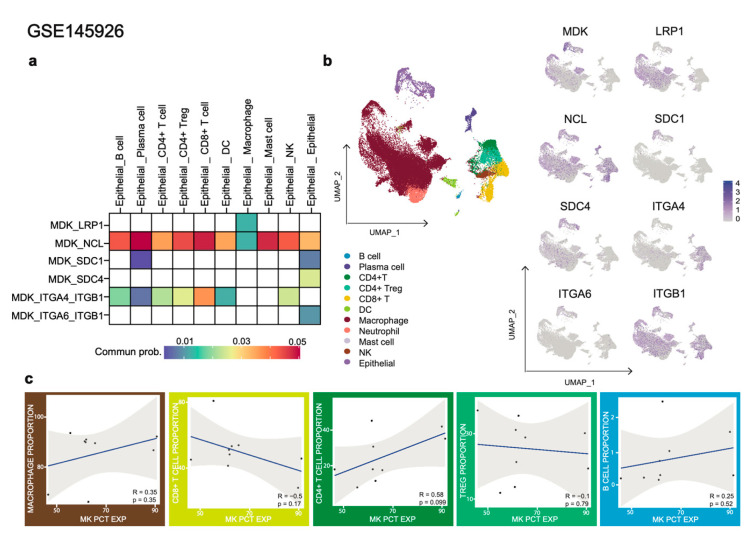
Analysis of COVID-19 bronchoalveolar scRNA-seq cohort. (**a**) Heatmap presents the ligand–receptor communication probability in BALF collected from COVID-19 patients (GSE145926, n = 9). Communication probability was visualized using the ggplot package. (**b**) Expression of cell type clustering (**left**) and ligand–receptor expression (**right**), presented as expression threshold values (key), overlaid on the t-SNE maps. (**c**) MK percentage expression (MK PCT EXP) (*x*-axis) plotted against the cell type abundance (*y*-axis), presented as a scatter plot: blue solid line, linear regression (least-square method) and Pearson correlation coefficient ‘R’ are displayed. Each dot represents an individual patient.

**Figure 5 cancers-15-04850-f005:**
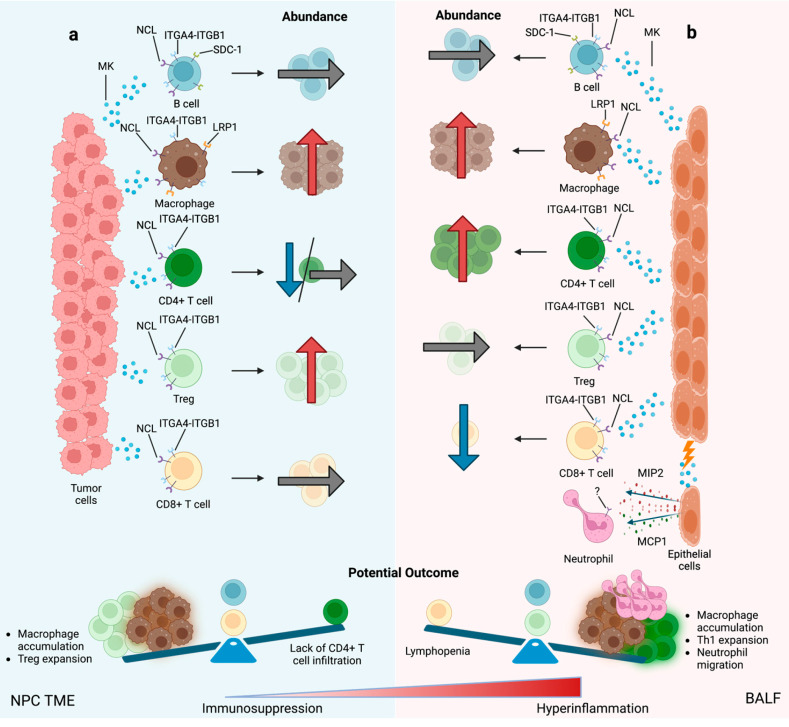
The impact of MK on cell and ligand–receptor pair abundance in disease microenvironments. Proposed MK-mediated cell–cell interactions, L-R pairs, MK-cytokine signaling networks in the microenvironment of NPC (**a**) and COVID-19 (**b**). Arrows indicate directionality interaction between MK expression and immune cells: positive (red), negative (blue), and no change (gray).

**Table 2 cancers-15-04850-t002:** Roles of MK in different cancer types.

Tumor Biological Roles	Indications	Signaling and Mechanisms	Responses Involved	Ref.
Immune suppression	Melanoma	NF-κB signaling activation, immunomodulator secretion	Educating macrophages towards immune tolerant states. Inducing resistance towards immune checkpoint blockade (ICB) through CD8+ T cells dysfunction	[33]
	Angiomyolipoma/Lymphangioleiomyomatosis (AML/LAM)	mTORC1 pathway	Immunosuppression and drug resistance	[59]
	Glioma	p53 from DNA damage repair alteration promote MK activation and overexpression	Facilitating glioma cells and microglia communication, lead to M2 polarization of microglia	[46]
Tumor growth, progression	HCC	PI3K/AKT/NF-κB/TrkB signaling	Intensifying anoikis resistance, supporting recurrence and metastasis	[69]
	HCC	STAT3 pathway activation via stimulating JAK1/2 phosphorylation	Tumor growth and invasion	[70]
	HCC, kidney and breast cancers	Disruption of LKB1-STRAD-Mo25 complex, leads to AMPK suppression	Cancer cell proliferation	[51]
	Neuroblastoma	Notch2-Hes-1 signaling	Tumorigenesis	[52]
	HNSCC	Tyrosine phosphorylation of FAK, paxillin and Stat1α pathway activation	Tumor cell migration and invasion	[39]
	NSCLC	NF-κB-Hes-1 signaling	Hypoxia-induced progression and metastasis	[49]
Angiogenesis	NSCLC	VEGF-induced angiogenesis	Endothelial cells migration and neovascularization	
	Breast cancer	Endothelial proliferation	Increasing vascular density	[53]
	HCC	Interaction with progranulin	Endothelial cells proliferation, migration, and tubulogenesis	[54]
Epithelial-mesenchymal transition (EMT)	NSCLC	Notch2-NF-κB-Hes-1 signaling	Upregulation of EMT markers	[49]
Mitogenesis	Kidney cancer	JAK/STAT pathway	Autocrine tumor cells proliferation	[55]
	Breast cancer	NF-κB-NR3C1 pathway	Tumor cells proliferation	[56]
Anti-apoptosis	Prostate cancer	Extracellular signal-regulated kinase 1/2 and p38 mitogen-activated protein kinase pathway	Suppression of TNF-α-induced apoptosis	[58]
	HCC	TRAIL/ActD-mediated pathway	Protect cells from caspase-3-mediated apoptosis	[57]
Drug resistance	Neuroblastoma and osteosarcoma	AKT pathway	Cytoprotection of neighboring drug-sensitive cells, contribute to chemotherapy resistance	[71]
	Prostate cancer	PI3K/AKT and MAPK/ERK pathway	Reduce quercetin efficacy on prostate cancer stem cell proliferation	[72]

**Table 3 cancers-15-04850-t003:** Patents filed on MK.

Patent Number	Disease/Process	Treatment	Company, Publish Date
WO2018016674A1	Brain cancer	An MK inhibitor to overcome resistance to temozolomide and/or radiation therapy	Samsung Electronic Co. Ltd., Suwon, South Korea, 2018
WO2016058047A1	Cancer, inflammatory and auto-immune diseases	Humanized antibody, IP14, to block MK	Cellmid Limited, Sydney, Australia, 2016
US8080649B2	Cancer, autoimmune disease, postoperative adhesion, and endometriosis	Aptamer possessing inhibitory activity against MK	Ribomic Inc, Tokyo, Japan, 2011
US20110159022A1	Cancers	Peptide derived from MK with CD4 T/CD8 T epitope restricted by HLA molecules, as anticancer vaccine or monitoring of cellular response against MK during cancer or anticancer treatment	Commissariat a lEnergie Atomique et aux Energies Alternatives, Paris, France, 2011
US9658233B1	Tumor growth diagnosis	A kit and method to determine MK and pleiotrophin level	Dept of Health and Human Services, Washington, DC, USA, 2017

## Data Availability

scRNA-seq results generated for this study have been deposited in Gene Expression Omnibus (GEO) under the following accession numbers: GSE150430 study (https://www.ncbi.nlm.nih.gov/geo/query/acc.cgi?acc=GSE150430 (accessed on 10 May 2023)); GSE162025 study (https://www.ncbi.nlm.nih.gov/geo/query/acc.cgi?acc=GSE162025 (accessed on 10 May 2023)); GSE145926 study (https://www.ncbi.nlm.nih.gov/geo/query/acc.cgi?acc=GSE145926 (accessed on 10 May 2023)).

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
