# Peer review of "Nasopharynx Battlefield: Cellular Immune Responses Mediated by Midkine in Nasopharyngeal Carcinoma and COVID-19"

_cancers, 2023, doi:10.3390/cancers15194850_

Round 1

Reviewer 1 Report

Overall, I found the article to be well written and structured. The review article provides with a thorough overview of MK and MK-mediated cellular immunity in NPC and COVID-19 infections. The discussion on the potential influence of MK-mediated cellular immunity on COVID-19-contracted NPC cancer progression and therapy response is particularly insightful. Additionally, the diagrams and tables presented in the article are clear and informative, enhancing the overall readability of the manuscript.

I have a few minor suggestions to improve the manuscript.

1. While the article is comprehensive, it tends to delve into excessive detail and references to well-established findings or genera facts. To avoid overwhelming readers, I recommend trimming down some of the descriptions and focusing on providing key takeaways. By doing so, the article will strike a better balance between providing necessary information and avoiding unnecessary repetition.

2. Regarding the title, the term “intercellular communication” can have two interpretations: communication between MK-released cells and receptor cells (i.e., immune cells), or communication between different immune cells that are regulated by MK. Given that the article predominantly focuses on MK-mediated immune response, it would be beneficial to reconsider the title to specifically indicate the MK-mediated immune response. This adjustment will ensure the title accurately reflects the main focus of the article. 

3. In the discussion of the three patient cases (Lines 494-511), it would be helpful to clarify the aim and takeaway of these cases. Currently, the purpose of discussing these cases is not explicitly stated, making it challenging for readers to understand the relevance and implications of the presented information.

Author Response

Manuscript ID Cancers - cancers-2485555, “Nasopharynx Battlefield: Cell-cell Communication by Midkine in Nasopharyngeal Carcinoma and COVID-19” by Kam et al.

We thank the Reviewer for their time and expert opinion in assessing our manuscript. We are grateful for the extremely positive comments and constructive criticism to our work. We have carefully addressed the Reviewer’s comments and revised our manuscript, while also performed English editing. We have included a point-by-point response to the Reviewers’ comments below. Modifications to the text can be followed in the highlighted version of the revised manuscript. 

Comments to the author:

Overall, I found the article to be well written and structured. The review article provides with a thorough overview of MK and MK-mediated cellular immunity in NPC and COVID-19 infections. The discussion on the potential influence of MK-mediated cellular immunity on COVID-19-contracted NPC cancer progression and therapy response is particularly insightful. Additionally, the diagrams and tables presented in the article are clear and informative, enhancing the overall readability of the manuscript.

I have a few minor suggestions to improve the manuscript.

  1. While the article is comprehensive, it tends to delve into excessive detail and references to well-established findings or genera facts. To avoid overwhelming readers, I recommend trimming down some of the descriptions and focusing on providing key takeaways. By doing so, the article will strike a better balance between providing necessary information and avoiding unnecessary repetition.

Our reply: We thank the Reviewer for this comment and agree that the excessive detail and information would be difficult for readers to comprehend. We are aware of the study message and agree that unnecessary repetition should be avoided.

We have now trimmed down some of the well-established findings or genera facts regarding the overview of Midkine (MK), as described in Section 2 of the original manuscript. In addition, we have also rephrased the subtitle in this section, which is now stated as “Midkine physiological and pathophysiological properties” (line 102, page 3). We sincerely hope that our revision achieves a better balance in understanding MK-mediated immune responses in the context of COVID-19-contracted NPC cancer progression and therapy response in the revised manuscript.

  1. Regarding the title, the term “intercellular communication” can have two interpretations: communication between MK-released cells and receptor cells (i.e., immune cells), or communication between different immune cells that are regulated by MK. Given that the article predominantly focuses on MK-mediated immune response, it would be beneficial to reconsider the title to specifically indicate the MK-mediated immune response. This adjustment will ensure the title accurately reflects the main focus of the article. 

Our reply: We thank the reviewer for pointing this out and we apologize if the wording of the title was lacking focus. We agree that it is important to rename the title in order to specifically indicate the MK-mediated immune response. We have now re-phrased the title in the main content for a better clarity. We copied here for the new title: “Nasopharynx Battlefield: Cellular Immune Responses Mediated by Midkine in Nasopharyngeal Carcinoma and COVID-19”.

  1. In the discussion of the three patient cases (Lines 494-511), it would be helpful to clarify the aim and takeaway of these cases. Currently, the purpose of discussing these cases is not explicitly stated, making it challenging for readers to understand the relevance and implications of the presented information.

Our reply: We thank the reviewer for highlighting this point which was not adequately presented. In this part of the review, we aim to present clinical evidences that patients with weakened immunity, such as those undergoing radio/chemotherapy as a first-line treatment like NPC patients, are more susceptible to coronavirus infection and can experience more severe effects. In our original manuscript, we did mention that "Despite the prevailing notion that patients with comorbidities like cancer and post-chemoradiation have weakened immunity, which can facilitate the transmission of coronavirus infection in these patients" (lines 512-514, page 16). However, we rectified that this information was not explicitly stated. To improve the clarity and readability of the section. We have now provided some description before discussing the patient cases in the revised manuscript, thus enhancing the relevance and implications of the presented cases. We copied here for the new description: “In this section, several patient cases presenting a disproportionate imbalance in immunity will be discussed, illustrating the significant challenges encountered by oncology practitioners when faced with the decision of treating a cancer patient with an ongoing antineoplastic treatment who has also been diagnosed with COVID-19.” (lines 485-488, page 15). We also feel it would be clearer to rephrase the subtitle of this section in order to clarify the aim and takeaway of these cases. We copied here for the new subtitle: “Managing NPC and COVID-19: Challenges and Clinical Evidence for Standard Care Therapies” (lines 481-482, page 15).

Thank you very much once again for your expert review.

Reviewer 2 Report

In this review, Kam et al. sumarize data on the recently dicovered cytokine midkine. They focus mainly on the effects of this peptide on nasopharyngeal carcinomas. In addition, they describe the effect this cytokine could have when a NPC patient is infected by SARS-COV-2. The latter part is of course still very speculative since data still very scarse. Nevertheless they present ideas that should be followed. The manuscript is well written, timely and comprehensive. I recommned publication as it is.

Author Response

Manuscript ID Cancers - cancers-2485555, “Nasopharynx Battlefield: Cell-cell Communication by Midkine in Nasopharyngeal Carcinoma and COVID-19” by Kam et al.

We thank the Reviewer for their time and expert opinion in assessing our manuscript. We are grateful for the extremely positive comments and constructive criticism to our work. We have carefully addressed the Reviewer’s comments and revised our manuscript, while also performed English editing. We have included a point-by-point response to the Reviewers’ comments below. Modifications to the text can be followed in the highlighted version of the revised manuscript. 

Comments to the author:

In this review, Kam et al. sumarize data on the recently dicovered cytokine midkine. They focus mainly on the effects of this peptide on nasopharyngeal carcinomas. In addition, they describe the effect this cytokine could have when a NPC patient is infected by SARS-COV-2. The latter part is of course still very speculative since data still very scarse. Nevertheless they present ideas that should be followed. The manuscript is well written, timely and comprehensive. I recommned publication as it is.

Our reply: Thank you so much for your appreciation. We hope that our work is educational and informative, as it covers various aspects of the midkine-mediated immune responses between NPC and COVID-19.

Reviewer 3 Report

Dear authors,

this review is very interesting and provides an overall view on the role that Sars-Cov-2 infection on the incidence and epidemiology of tumors.

The figures are well made and educational.

I think that you can inprove the role of other co-factors in Nasopharyngeal Carcinoma, as EBV infection, both in introduction and discussionj section.

Author Response

Manuscript ID Cancers - cancers-2485555, “Nasopharynx Battlefield: Cell-cell Communication by Midkine in Nasopharyngeal Carcinoma and COVID-19” by Kam et al.

We thank the Reviewer for their time and expert opinion in assessing our manuscript. We are grateful for the extremely positive comments and constructive criticism to our work. We have carefully addressed the Reviewer’s comments and revised our manuscript, while also performed English editing. We have included a point-by-point response to the Reviewers’ comments below. Modifications to the text can be followed in the highlighted version of the manuscript. 

Comments to the author:

Dear authors,

this review is very interesting and provides an overall view on the role that Sars-Cov-2 infection on the incidence and epidemiology of tumors.

The figures are well made and educational.

I think that you can inprove the role of other co-factors in Nasopharyngeal Carcinoma, as EBV infection, both in introduction and discussionj section.

Our reply: Thank you very much for your nice comments. Following the Reviewer’s suggestion, we have now elaborated further on the co-factors in NPC in the relevant sections. Besides adding the long-known signature role of EBV in NPC pathogenesis in the introduction section (lines 58-63, page 2), it is worth discussing the current challenges of NPC clinical research on various immunotherapeutic approaches, including EBV specific monoclonal antibodies. This discussion can guide further research in this direction.  We feel that it would be appropriate to add this new information into the discussion section (lines 625-636, pages 18-19) without detriment to the overall message of the revised manuscript.

Thank you very much once again for your expert review.

Reviewer 4 Report

Thank you so much for undertaking the huge task of writing this review article summarizing the results of your own and those published in the literature for the role of Midkine (MK) in different cancer types, specifically for Nasopharyngeal Carcinoma, and COVID-19. It included a thorough review of many aspects of MK and discussed its therapeutic implications. I have two suggestions:

1. The article is very long and difficult for readers to comprehend. Please add a paragraph at the end of the introduction section to give an overview and the structure of all sections that will be included.

2. Since the focus (and the title) is on the implication of MK in NPC patients infected with COVID-19, the conclusion can be expanded to address the question raised at the beginning of this article: that is "questions that experts are trying to answer about COVID-19 is whether it will have a positive or negative effect on cancer", and focus on NPC.

There were some minor grammar errors, which can be improved by carefully reading and editing the article again.

Author Response

Manuscript ID Cancers - cancers-2485555, “Nasopharynx Battlefield: Cell-cell Communication by Midkine in Nasopharyngeal Carcinoma and COVID-19” by Kam et al.

We thank the Reviewer for their time and expert opinion in assessing our manuscript. We are grateful for the extremely positive comments and constructive criticism to our work. We have carefully addressed the Reviewer’s comments and revised our manuscript, while also performed English editing. We have included a point-by-point response to the Reviewers’ comments below. Modifications to the text can be followed in the highlighted version of the manuscript. 

Comments to the author:

Thank you so much for undertaking the huge task of writing this review article summarizing the results of your own and those published in the literature for the role of Midkine (MK) in different cancer types, specifically for Nasopharyngeal Carcinoma, and COVID-19. It included a thorough review of many aspects of MK and discussed its therapeutic implications. I have two suggestions:

  1. The article is very long and difficult for readers to comprehend. Please add a paragraph at the end of the introduction section to give an overview and the structure of all sections that will be included.

Our reply: We thank the Reviewer for this comment. We apologise if the article tends into difficult to comprehend. We are aware of the study message and feel that it would be appropriate to achieve a better balance in understanding MK-mediated immune responses in the context of COVID-19-contracted NPC cancer progression and therapy response.  Therefore, we have now trimmed down some of the well-established findings or genera facts regarding the overview of Midkine (MK), as described in Section 2 of the original manuscript. In addition, we have also rephrased the subtitle in this section, which is now stated as “Midkine physiological and pathophysiological properties” (line 102, page 3).

In our original manuscript, we have mentioned what would be discussed in the preceding sections of the review at the end of the introduction.  We apologise if the description of the review’s structure was not clear.  As suggested by the Reviewer, we have now revised the description to provide better clarity on the structure of all sections that will be included (lines 87-100, page 3). We copied here for the new description: “This article aims to comprehensively detail the MK-mediated cellular immunity that may impact immunosurveillance mechanisms in both of these nasal-related diseases. The review will begin by summarizing the physiological and pathophysiological properties of MK. We will then highlight the potential role of MK in contributing to carcinogenesis and cancer progression, with a specific focus on intercellular communication mediated by MK in the tumor microenvironment (TME) of NPC. The coordinated gene expression of ligands and receptors will be discussed, along with its implications for inferring intercellular communication, and new ideas for future application prospects will be put forward. In the context of the relationship between COVID-19 and NPC, possible cell-cell communication involving MK to regulate COVID-19 disease will be discussed. Additionally, current cases of NPC patients who have contracted COVID-19, leading to a significant interruption of cancer research, will be outlined. Finally, ongoing therapeutic strategies aimed at controlling the onset or damage caused by the overproduction of MK will be summarized, with the goal of reducing both morbidity and mortality.”

  1. Since the focus (and the title) is on the implication of MK in NPC patients infected with COVID-19, the conclusion can be expanded to address the question raised at the beginning of this article: that is "questions that experts are trying to answer about COVID-19 is whether it will have a positive or negative effect on cancer", and focus on NPC.

Our reply: We thank the Reviewer for this valuable comment and we agree with this reviewer that the conclusion should be expanded to address the question raised in the summary of this article. We have further elaborated on this in the revised manuscript (lines 607-624, page 18). We copied here for the new information: “To summarize, COVID-19 and cancer, despite having discrete etiologies, both involve an imbalance in the immune cells compartment. The depletion of CD8 T cells, which are essential for immune response, by COVID-19 may aggravate cancer progression in NPC patients, suggesting a potentially negative impact. Furthermore, the interaction between immunomodulatory molecules and immune cells during COVID-19 infection may further disturb the already compromised immune system in NPC, exacerbating the negative effects. On the other hand, in some cases, if COVID-19 triggers a robust immune response of NPC patients, it may lead to improved tumor surveillance and enhanced immune recognition of cancer cells. This introduces uncertainties regarding the directional effects of COVID-19 on NPC, as the impact may vary depending on different contexts. As discussed in this review, MK is involved in a multitude of biological and cell-cell communication processes, and is believed to play parallel roles in the mechanisms of immunosuppressive cancers and the COVID-19 cytokine storm. As such, MK could be a useful "soldier" to fight against the early phases of cancers and COVID-19 infections, which are dominated by the innate immune response. However, as the diseases progress, MK may become a driving force of dysregulation in the local inflammatory milieu. This highlights the complex and context-dependent nature of the interplay between COVID-19, cancer, and the role of specific immunomodulatory molecules like MK.”

We sincerely hope that our revision achieves a better conclusion in understanding MK-mediated immune effect in the context of COVID-19-contracted NPC cancer progression and therapy response in the revised manuscript. We have also performed English editing of our manuscript, and hope that it is now more comprehensible.

Thank you very much once again for your expert review.

Round 2

Reviewer 3 Report

I' m agree for paper pubblication.